# Wave Characteristics of Coagulation Bath in Dry-Jet Wet-Spinning Process for Polyacrylonitrile Fiber Production Using Computational Fluid Dynamics

**Son Ich Ngo** [1], **Young-Il Lim** [1,*] **and Soo-Chan Kim** [2]

1 Center of Sustainable Process Engineering (CoSPE), Department of Chemical Engineering, Hankyong National University, Jungang-ro 327, Anseong-si 17579, Korea; ngoichson@hknu.ac.kr
2 Research Center for Applied Human Sciences, Department of Electrical and Electronic Engineering, Hankyong National University, Jungang-ro 327, Anseong-si 17579, Korea; sckim@hknu.ac.kr
* Correspondence: limyi@hknu.ac.kr; Tel.: +82-31-670-5207; Fax: +82-31-670-5209

**Abstract:** In this work, a three-dimensional volume-of-fluid computational fluid dynamics (VOF-CFD) model was developed for a coagulation bath of the dry-jet wet spinning (DJWS) process for the production of polyacrylonitrile (PAN)-based carbon fiber under long-term operating conditions. The PAN-fiber was assumed to be a deformable porous zone with variations in moving speed, porosity, and permeability. The Froude number, interpreted as the wave-making resistance on the liquid surface, was analyzed according to the PAN-fiber wind-up speed ($v_{PAN}$). The effect of the PAN speed on the reflection and wake flow formed by drag between a moving object and fluid is presented. A method for tracking the wave amplitude with time is proposed based on the iso-surface of the liquid volume fraction of 0.95. The wave signal for 30 min was divided into the initial and resonance states that were distinguished at 8 min. The maximum wave amplitude was less than 0.5 mm around the PAN-fiber inlet nozzle for $v_{PAN}$ = 0.1–0.5 m/s in the resonance state. The VOF-CFD model is useful in determining the maximum $v_{PAN}$ under an allowable air gap of the DJWS process.

**Keywords:** polyacrylonitrile-based carbon fiber; coagulation bath; dry-jet wet spinning process; computational fluid dynamics; wave resonance; maximum wave amplitude

---

## 1. Introduction

Carbon fiber (CF) has attracted attention owing to the increase of its uses in the fields of aerospace, automotives, sporting goods, biomedicine, building, and infrastructure [1–4]. The need for high-quality and high-productivity CF has emerged due to a need for economic efficiency [5,6]. Polyacrylonitrile (PAN)-based carbon fibers constitute an overwhelming share, i.e., more than 90%, of the world's CF production [7]; they are advantageous in terms of their high tensile strength, low density, and reasonable cost [8].

Several spinning processes such as wet, dry, dry-jet wet, and melt spinning are used to fabricate PAN-fiber [7,9,10]. The dry-jet wet spinning (DJWS) process possesses the advantages of high-speed fiber formation, high concentration of dope, high degree of jet stretch, and control capability of coagulation kinetics [9,11]. The DJWS process is characterized by the fact that the fiber solution is extruded into air or a gaseous environment, and is pulled inside a coagulation bath [11,12].

The gap between the spinneret and the coagulation bath surface, called the air gap, varies with the type of polymer and technology being used. The air gap does not only facilitate jet-stretch, but also provides resistance to counter diffusion when the dope is immersed inside the coagulation bath; this is a crucial factor for preventing the void structure of the fiber product [13]. By increasing the air gap

height, the maximum attainable spinning speed decreases sharply [11]. A large air gap results in a yarn of low tenacity [11] and high elongation stress [13]. A small air gap can lead to fiber breakage by contact between the spinneret nozzle and the coagulation bath surface. It has been reported that the molecular orientation of fibers induced by shear stress within the spinneret can relax in an air gap of 1 cm [14]. A high-speed PAN-fiber production has been presented using a solvent-free DJWS process [7].

Computational fluid dynamics (CFD) has been shown as a powerful tool for analyzing single-phase hydrodynamics [2,3] as well as multiphase interactions [15–18]. The volume-of-fluid (VOF) CFD for multiphase flows is widely used for capturing interphase surface characteristics such as tracking the interface between gas and liquid phases in structured packing [19], and free-surface wave flow around a surface-piercing foil [20]. The VOF method can handle highly distorted or breaking interfaces without the need for a grid conformation [18–20]. However, few researchers have addressed the VOF-CFD model the wave resonance on a free surface in the presence of obstacles and moving objects. Moreover, the detection of wave amplitude is required to identify an allowable air gap in the DJWS process.

The purpose of this study was to develop a VOF-CFD model of a coagulation bath in the DJWS process for the identification of wave resonance and amplitude on the liquid surface during a long-term operating condition. The coagulation bath geometry for a DJWS process, CFD mesh structure, and boundary conditions are presented. A method for tracking the wave amplitude with time is proposed based on the iso-surface of the liquid volume fraction. The Froude number, wave speed, and maximum wave amplitude (MWA) are investigated according to the PAN-fiber spinning speed using the VOF-CFD model.

## 2. Coagulation Bath of DJWS Process

The symmetric coagulation bath of a DJWS process adopted from a patent [21] is shown in Figure 1. The coagulation bath is 1400 mm × 300 mm × 1310 mm (length × width × height). The diameter of the spinneret nozzle is 120 mm (see Figure 1a). The PAN solution is extruded through a commercial spinneret nozzle with 3000 holes of 0.5 mm in each, and subsequently passes through the air gap before being immersed into a dimethyl sulfoxide (DMSO)–water solution in the coagulation bath. The as-spun fiber passes through a static guide roller immersed in the coagulation bath. In this CFD study, the PAN solution uniformly injected from the nozzle is assumed as a porous medium. The non-slip boundary condition was applied to the surface of the roller and the wall of the bath.

During the coagulation process, the PAN solution is spun, stretched, and wound to create the fiber. As shown in Figure 1b,c, the shape of the PAN-bundle (or PAN solution) deforms from a circle to an ellipse, and its size reduces linearly. The ellipse maintains the same form from A to B. The large ellipse shrinks linearly to a small one from B to C. It is known that a short air gap benefits jet-stretch and high production speed [11]. The DJWS process is operated for 30 min in the flow time.

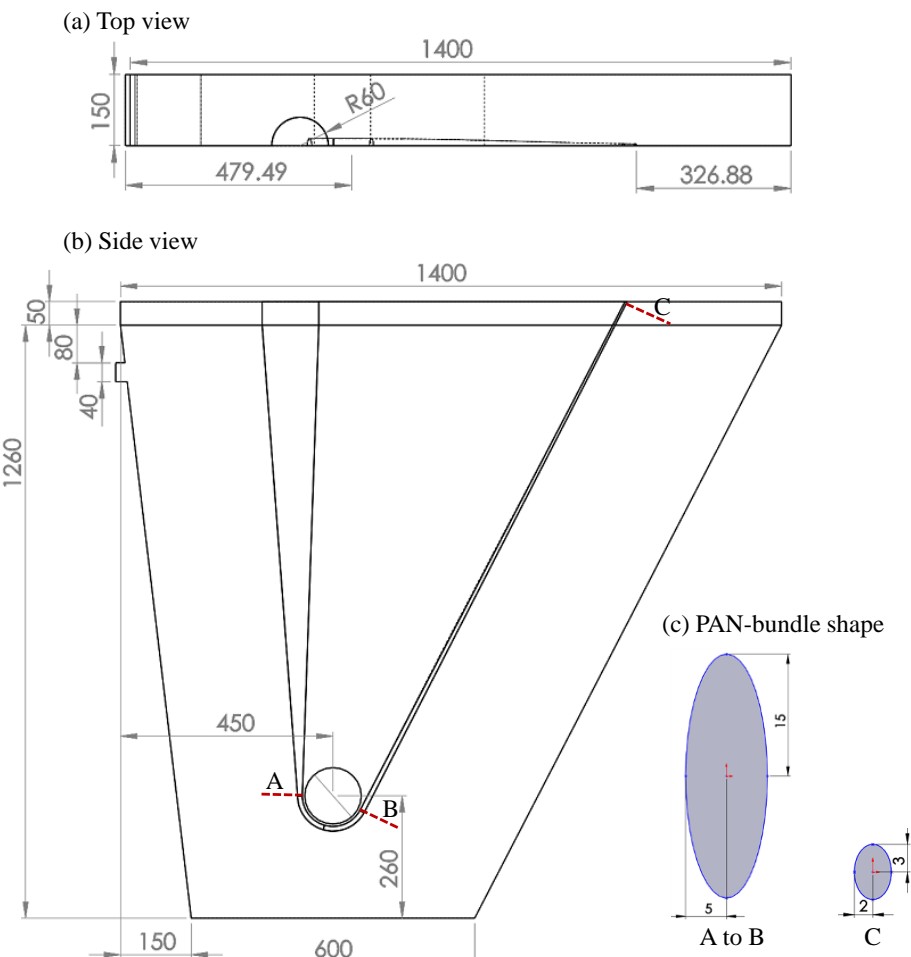

**Figure 1.** Geometry of coagulation bath and PAN bundle shape in dry-jet wet spinning (DJWS) process.

## 2.1. Mesh Structure and Material Properties

The boundary and mesh structure of the CFD domain are shown in Figure 2a. The inlet of the DMSO–water solution is located below the liquid outlet at the top of the bath. The liquid overflows from the liquid outlet. The PAN solution is injected 50 mm above the liquid surface. The PAN fiber solidified in the coagulation bath exits at the end of the bath.

The CFD domain has a polyhedral mesh structure of 0.67 million cells. The mesh is concentrated on the gas–liquid interface, PAN fiber, and walls. The near-PAN zone from the PAN inlet to the middle roller where a strong perturbation of liquid can occur is compartmentalized and a relatively dense mesh is used. The area at the top of the coagulation bath is named the freeboard.

Because the diameter and shape of the PAN fiber change and the PAN fiber is stretched through the roller, the PAN fiber is assumed to be a continuous porous zone with variable moving velocity ($v$), porosity ($\phi$), and permeability ($K$). As shown in Figure 2b, the speed of the PAN fiber is divided into four zones near the middle roller: $v^{in}$, $v_1^{mid}$, $v_2^{mid}$, and $v^{out}$. The first speed ($v^{in}$) and the last speed ($v^{out}$) are regarded as the PAN-fiber spinning speed ($v_s$) and wind-up speed ($v_{PAN}$), respectively.

The one-filament diameters of the PAN-bundle are $d_f^{in} = 0.5$ (as the spinneret hole diameter), $d_{f,1}^{mid} = d_{f,2}^{mid} = 0.2$, and $d_f^{out} = 0.1$ mm [7]. The local porosity ($\phi$) of the porous zone is calculated from

the ratio of the cross-sectional area of all the filaments to that of the PAN bundle. Thus, $\phi$ is defined as the reciprocal of a linear function of the $Y$-coordinate, i.e.,

$$\begin{cases} \frac{1}{\phi^{in}} = 0.9479 - 4.79 \times 10^{-2} \times (Y - 0.26) \\ \frac{1}{\phi^{mid}_{1,2}} = 0.9 - 1.61 \times (Y + 0.09449) \\ \frac{1}{\phi^{out}} = 0.6875 \end{cases} \tag{1}$$

where the origin of $Y$ is the bottom of the coagulation bath.

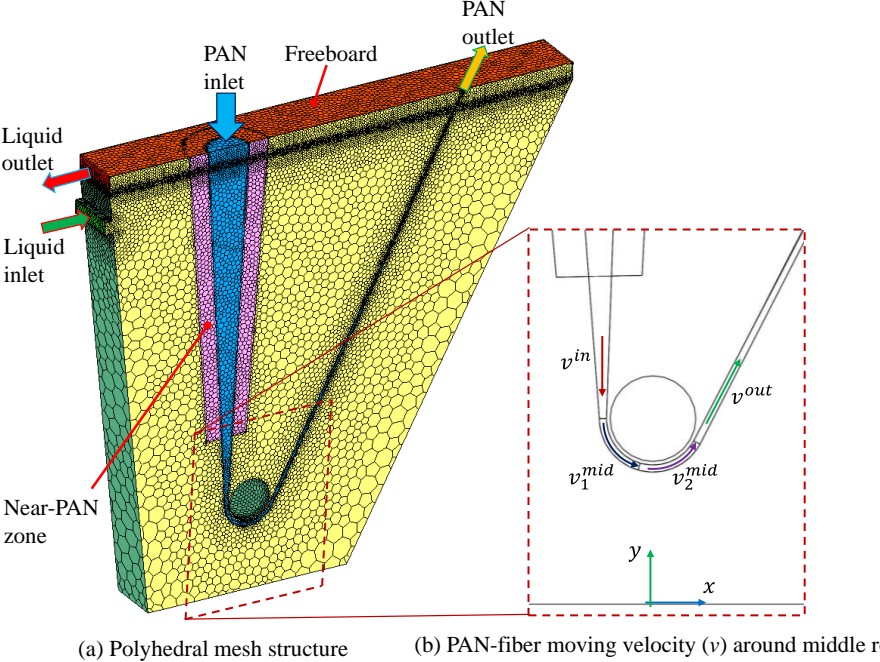

(a) Polyhedral mesh structure          (b) PAN-fiber moving velocity ($v$) around middle roller

**Figure 2.** Mesh structure and polyacrylonitrile (PAN)-fiber moving velocity.

The transverse and parallel permeabilities ($K_\perp$ and $K_\parallel$, respectively) are calculated from [22,23]

$$K_\perp = \frac{16}{9\pi\sqrt{6}} \left( \sqrt{\frac{1-\phi_c}{1-\phi}} - 1 \right)^{2.5} R_f^2 \tag{2}$$

$$K_\parallel = \frac{8R_f^2}{c} \frac{\phi^3}{(1-\phi)^2} \tag{3}$$

where $\phi_c = 1 - \pi/\left(2\sqrt{3}\right)$ is the critical porosity for hexagonal packing, at which the filaments come into contact, thus preventing flow in the transverse direction and $c$ is the geometrical shape factor equal to 53 for hexagonal packing. $R_f$ is the filament radius (= $d_f/2$). The porous viscous resistance ($1/K$) is the reciprocal of the permeability. $K_\perp$ and $K_\parallel$ along the $Y$-axis are defined as

$$\begin{cases} \frac{1}{K_\perp^{in}} = 3.86 \times 10^6 + 7.15 \times 10^7 (Y - 0.26) \\ \frac{1}{K_{1,2,\perp}^{mid}} = 7.15 \times 10^7 + 3.1 \times 10^{10} (Y + 0.09449) \\ \frac{1}{K_\perp^{out}} = 4.17 \times 10^9 \end{cases} \tag{4}$$

$$\begin{cases} \frac{1}{K_{\parallel}^{in}} = 3.38E5 + 8.75 \times 10^6 \times (Y - 0.26) \\ \frac{1}{K_{1,2,\parallel}^{mid}} = 9.09E6 + 5.96 \times 10^9 \times (Y + 0.09449) \\ \frac{1}{K_{\parallel}^{out}} = 7.96 \times 10^8 \end{cases} \tag{5}$$

In this study, five spinning velocities ($v_s$) of the dope (polymer solution) at the spinneret nozzle were considered with a jet-stretch ratio ($v_{PAN}/v_s$) of five [7,24,25]. Table 1 indicates the PAN-bundle velocity ($v$), porosity ($\phi$), and viscous resistance ($1/K$) according to the five cases. The porous zone was divided into four: the inlet, middle roller 1, middle roller 2, and outlet. The five $v_{PAN}$ were 0.1, 0.25, 0.5, 0.75, and 1 m/s. $v_{PAN}$ was five times higher than $v_s$ according to the jet-stretch ratio, implying that the PAN-fiber was elongated by five times during the coagulation. The ranges of porosity and viscous resistance were indicated in each porous zone.

**Table 1.** Material properties of porous media zones.

| Porous Zone | PAN-Bundle Velocity ($v$, m/s) | | | | | Porosity ($\phi$) | Viscous Resistance ($1/K$, m$^2$) |
| --- | --- | --- | --- | --- | --- | --- | --- |
| | Case 1 | Case 2 | Case 3 | Case 4 | Case 5 | | |
| Inlet | 0.02 | 0.05 | 0.1 | 0.15 | 0.2 | 0.9479–0.9 | $3.86 \times 10^6$–$7.54 \times 10^7$ |
| Middle roller 1 | 0.03 | 0.075 | 0.15 | 0.225 | 0.3 | 0.9 | $7.54 \times 10^7$ |
| Middle roller 2 | 0.04 | 0.1 | 0.2 | 0.3 | 0.4 | 0.9–0.6875 | $7.54 \times 10^7$–$4.17 \times 10^9$ |
| Outlet | 0.1 | 0.25 | 0.5 | 0.75 | 1 | 0.6875 | $4.17 \times 10^9$ |

### 2.2. Mesh-Independence Test

A mesh-independence test was performed for Case 2 ($v_{PAN}$ = 0.25 m/s) on coarse, medium, and fine meshes of 0.52, 0.673, and 1.42 million cells, respectively. The time- and volume-averaged velocities in the near-PAN zone and the entire CFD domain are shown in Figure 3 with respect to the cell number. The average velocities of the coarse and medium meshes were significantly different, whereas those on the medium and fine meshes changed slightly. Therefore, the medium mesh was selected for computational efficiency, while maintaining the numerical accuracy.

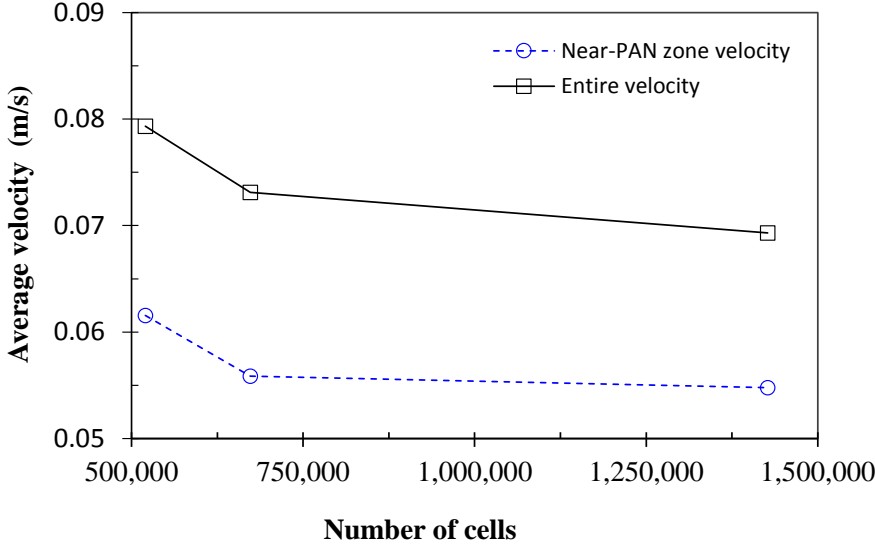

**Figure 3.** Mesh-independence test of computational fluid dynamics (CFD) domain at $v_{PAN}$ = 0.25 m/s (Case 2).

It is noteworthy that the mesh density at the liquid surface should be sufficiently dense to capture the surface wave. We recommend that the cell size at the liquid surface be smaller than 0.5 mm,

according to our experience. The cell size at the liquid surface ranges from 0.38 to 0.45 mm on the selected medium mesh.

## 3. VOF-CFD Model

The volume-of-fluid computational fluid dynamics model relies on the assumption that two phases do not interpenetrate [2]. An unsteady-state VOF-CFD model was applied to the incompressible gas–liquid system including air and the DMSO–water solution, where the diffusion of the solvent (DMSO) from the PAN solution to the bath was ignored. The standard $k$–$\varepsilon$ model was used for modeling the turbulence, which has been used extensively for industrial applications [19]. Table 2 shows the VOF-CFD model.

**Table 2.** VOF-CFD model used in this study.

| | |
|---|---|
| Continuity equation (DMSO): $\frac{\partial \alpha_D}{\partial t} + \vec{\nabla} \cdot (\alpha_D \vec{u}) = 0$ | (T1) |
| (air): $\alpha_D + \alpha_A = 1$ | (T2) |
| Single momentum equation: $\frac{\partial}{\partial t}(\rho \vec{u}) + \vec{\nabla} \cdot (\rho \vec{u} \vec{u}) = -\vec{\nabla} P + \vec{\nabla} \cdot [\eta(\vec{\nabla}\vec{u} + \vec{\nabla}\vec{u}^T)] + \rho \vec{g} + F_{surf}$ | (T3) |
| where the surface tension force ($F_{surf}$) is $F_{surf} = \vec{\nabla} \cdot \overline{\overline{\tau}}$ | (T4) |
| with the surface stress tensor of $\overline{\overline{\tau}} = \sigma \left( |\vec{\nabla}\alpha_D|\overline{\overline{I}} - \frac{\vec{\nabla}\alpha_D \cdot (\vec{\nabla}\alpha_D)^T}{|\vec{\nabla}\alpha_D|} \right)$ | (T5) |
| Single properties (density): $\rho = \alpha_A \rho_A + \alpha_D \rho_D$ | (T6) |
| (viscosity): $\eta = \alpha_A \eta_A + \alpha_D \eta_D$ | (T7) |
| $k$-$\varepsilon$ turbulence model: $\frac{\partial}{\partial t}(\rho k) + \vec{\nabla} \cdot (\rho k \vec{u}) = \vec{\nabla} \cdot \left[ \left( \eta + \frac{\eta_t}{D_k^t} \right) \vec{\nabla}k \right] + G_k - \rho\varepsilon$ | (T8) |
| $\frac{\partial}{\partial t}(\rho\varepsilon) + \vec{\nabla} \cdot (\rho\varepsilon\vec{u}) = \vec{\nabla} \cdot \left[ \left( \eta + \frac{\eta_t}{D_\varepsilon^t} \right) \vec{\nabla}\varepsilon \right] + C_{1\varepsilon}\frac{\varepsilon}{k}\left( G_k - C_{2\varepsilon}\rho\frac{\varepsilon^2}{k} \right)$ | (T9) |
| where: $\eta_t = \rho C_\mu \frac{k^2}{\varepsilon}$, $G_k = \eta_t S^2$, and $S = \sqrt{2S_{ij}Sij}$. | (T10) |

The tracking of the interface between the two phases was accomplished by the solution of a continuity equation for the volume fraction ($\alpha_D$) of the DMSO solution, where the DMSO solution is denoted as the subscript $D$. Assuming no mass transfer between the two phases and no source term in each phase, the continuity equation of the DMSO solution is expressed in Equation (T1), where $\vec{u}$ is the velocity. The continuity equation of the air phase (denoted as the subscript $A$) is computed using Equation (T2).

In this study, a special interpolation treatment of $\alpha_D$ to the cells that lie near the interface between the two phases was applied, which was based on the compressive scheme of a second-order slope limiter reconstruction [26]. To specify a cutoff limit for $\alpha_D$, a cutoff factor of $\alpha_D = 1 \times 10^{-6}$ was used. Thus, all values of $\alpha_D$ below the cutoff value were zero.

In the VOF model, the single momentum equation shown in Equation (T3) was solved throughout the domain, and the resulting velocity field was shared in both phases. The left-hand side of Equation (T3) includes the accumulation and convection of momentum per unit volume. On the right-hand side of Equation (T3), $P$ is the pressure shared in both phases, $\rho \vec{g}$ is the gravity force, $\eta(\vec{\nabla}\vec{u} + \vec{\nabla}\vec{u}^T)$ is the stress tensor of the Newtonian fluid, and $F_{surf}$ is the surface tension force on the interface between the two phases.

The continuum surface stress (CSS) model, expressed in Equation (T4), was used to model $F_{surf}$ conservatively, where the surface stress tensor ($\overline{\overline{\tau}}$) owing to the surface tension is expressed in Equation (T5). The gradient of $\alpha_D$ ($\vec{\nabla}\alpha_D$) is the surface normal vector and $\overline{\overline{I}}$ is the unit tensor. The surface tension coefficient ($\sigma$) is assumed as a constant $\sigma = 0.072$ N/m, which is comparable to the air-water system [19].

The single properties of density ($\rho$) and viscosity ($\eta$) are employed to solve the VOF model, as expressed in Equations (T6)–(T7). For the DMSO–water solution, $\rho_D = 1048$ kg/m$^3$ and $\eta_D = 0.00212$ kg/m/s. For air, $\rho_A = 1.225$ kg/m$^3$ and $\eta_A = 1.7894 \times 10^{-5}$ kg/m/s.

The standard $k$-$\varepsilon$ models [27] for turbulence kinetic energy ($k$) and turbulence energy dissipation ($\varepsilon$) are expressed in Equations (T8) and (T9), respectively. The turbulence viscosity ($\eta_t$) is determined by the local values of $\rho$, $k$, $\varepsilon$, and a constant $C_\mu$ (see Equation (T10)). $G_k$ representing the generation of turbulence kinetic energy is estimated by the turbulence viscosity ($\eta_t$) and the modulus of mean rate-of-strain tensor ($S$). $D_k^t$ and $D_\varepsilon^t$ in Equations (T8) and (T9), respectively, represent the diffusion rate of $k$ and $\varepsilon$, respectively [27]. The model constants of the turbulence model were set to $C_{1\varepsilon} = 1.44$, $C_{2\varepsilon} = 1.92$, $C_\mu = 0.09$, $D_k^t = 1.0$, and $D_\varepsilon^t = 1.3$.

### 3.1. Boundary Conditions of the VOF–CFD Model

Table 3 summarizes the boundary conditions for the present CFD model. The DMSO inlet was set to the mass flow inlet with 0.1 kg/s of the DMSO–water solution. The inlet and outlet of the PAN fiber, the outlet of the DMSO solution, and the freeboard (see Figure 2) were defined as an open channel. The open channel involved a free surface between the flowing fluid and the atmosphere that was often applied for wave propagation (ANSYS Fluent Theory Guide, ANSYS Inc., Washington, PA, USA, 2018).

**Table 3.** Boundary conditions of coagulation bath CFD simulation. Legend: DMSO, dimethyl sulfoxide.

| Boundary Type | Setting | Value | Remarks |
| --- | --- | --- | --- |
| DMSO inlet | Mass flow inlet | 0.1 kg/s | 100% liquid |
| PAN inlet | Open channel | $v_s = 0.02$–0.2 m/s | Porous medium |
| PAN outlet | Open channel | $v_{PAN} = 0.1$–1.0 m/s | Porous medium |
| DMSO outlet | Open channel | | |
| Freeboard | Open channel | | Gas phase |

### 3.2. Froude Number and Wave Speed

The open-channel flows are characterized by the dimensionless Froude Number ($Fr$) that is defined as the ratio of the inertial force to the hydrostatic force, i.e.,

$$Fr = \frac{\left|\vec{u}\right|}{\sqrt{gy}} \tag{6}$$

where $\left|\vec{u}\right|$ is the velocity magnitude of the liquid at the surface, $g$ is the gravity, and $y$ is the length scale. Here, $y$ is given by the distance from the bottom of the equipment to the free surface (approximately 1.26 m). The denominator of Equation (6) is the wave speed of the fluid itself.

The wave speed observed by a fixed observer is defined as (ANSYS Fluent Theory Guide, 2018):

$$\left|\vec{u}\right|_w = \left|\vec{u}\right| \pm \sqrt{gy} \tag{7}$$

Based on the Froude number, open-channel flows can be classified into the following three categories. When $Fr < 1$, i.e., $\left|\vec{u}\right| < \sqrt{gy}$, $\left|\vec{u}\right|_w < 0$ or $\left|\vec{u}\right|_w > 0$, and the flow is known to be subcritical where disturbances can travel to the upstream and downstream. In this case, the downstream conditions might influence the upstream flow. When $Fr = 1$, the flow is known to be critical, where waves propagating from the inlet stream remain stationary. When $Fr > 1$, i.e., $\left|\vec{u}\right| > \sqrt{gy}$, $\left|\vec{u}\right|_w > 0$, and the flow is known to be supercritical where disturbances cannot travel to the upstream (ANSYS Fluent Theory Guide, 2018).

## 4. Detection of Surface Wave

In the VOF multiphase CFD model, the interface between the gas and liquid is located at a liquid volume fraction from 0 to 1 ($0 < \alpha_D < 1$). $\alpha_D$ close to 0 is gas-like while $\alpha_D$ close to 1 is liquid-like. The interface sharpness was highly dependent on the grid resolution around the interface [20]. In this study, $\alpha_{D,min} = 0.95$ as the minimum liquid volume fraction was selected to detect the iso-surface of the liquid. $\alpha_{D,min}$ higher than 0.95 on the medium mesh structure yielded a flatter iso-surface. $\alpha_{D,min}$ lower than 0.95 caused a wave signal with considerable noise. A $\alpha_{D,min}$ lower than 0.95 was reported to capture the interface for the Eulerian multiphase CFD model [28,29].

The detection point of the surface wave is shown in Figure 4. There are five horizontal lines (1–5), six vertical lines (A–F), and six additional points (PA–PF) located near the PAN injection spinneret nozzle. Lines 1 and 5 are located at the symmetry line and the side wall of the coagulation bath, respectively. The distance from Lines 2 to 5 from the symmetry line (Line 1) is 20, 50, 100, and 149 mm, respectively. The vertical lines are positioned in the same interval from the spinneret nozzle (Line B) of 200 mm. Thus, 30 intercepts of the horizontal and vertical lines are created, which are named after the number and letter.

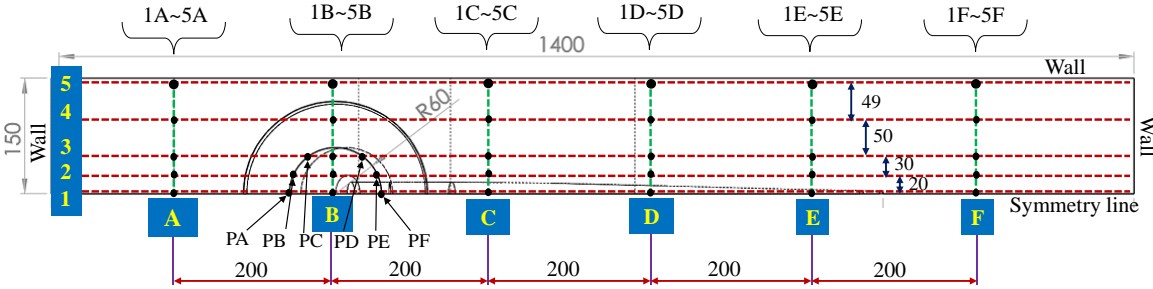

**Figure 4.** Detection points of surface wave.

At the 36 points, the surface wave signal in the time-series was transformed into the wave height ($h_w$) by a three-step data processing, as shown in Figure 5. In the first step, the unsteady state CFD simulation results were used to reconstruct the iso-surface of $\alpha_{D,min} = 0.95$. The data sampling interval was set to every one second of flow time, and 1800 CFD data files for 30 min were used. The wave coordinates ($Y^i$) at a point (P) and a time step ($t^i$) were obtained from the iso-surface of liquid with $\alpha_D \geq 0.95$. In the second step, the coordinate and time of the iso-surface, P($Y^i$), were exported as a new dataset. In the third step, the wave height ($h_w$) in the time-series at each detecting point was generated. The maximum wave amplitude (MWA) was calculated from the difference between the maximum and minimum values of $h_w$ within a given period.

$$\text{MWA} = \max{[h_w(t)]} - \min{[h_w(t)]} \tag{8}$$

As shown in the third step of Figure 5, the wave height signal is divided into two states: initial wave and resonance wave. The initial wave appears at the first moment of operation, where the wave is unstable due to the inertia force of liquid itself against the moving PAN fiber. For all five cases with different PAN speeds, the initial wave appeared when the flow time was less than 8 min. The resonance wave occurred in the later moment, showing a stable hydrodynamic state with the compromise of inertia and gravity forces, surface tension, and turbulence at the liquid surface.

In this study, the wave speed ($\left|\vec{u}\right|_w$) was calculated from two datasets of the iso-surface of liquid in an interval of 1 s. The distance of two wave peaks ($\Delta l$) displacing for one second was used directly as $\left|\vec{u}\right|_w$:

$$\left|\vec{u}\right|_w = \Delta l, \tag{9}$$

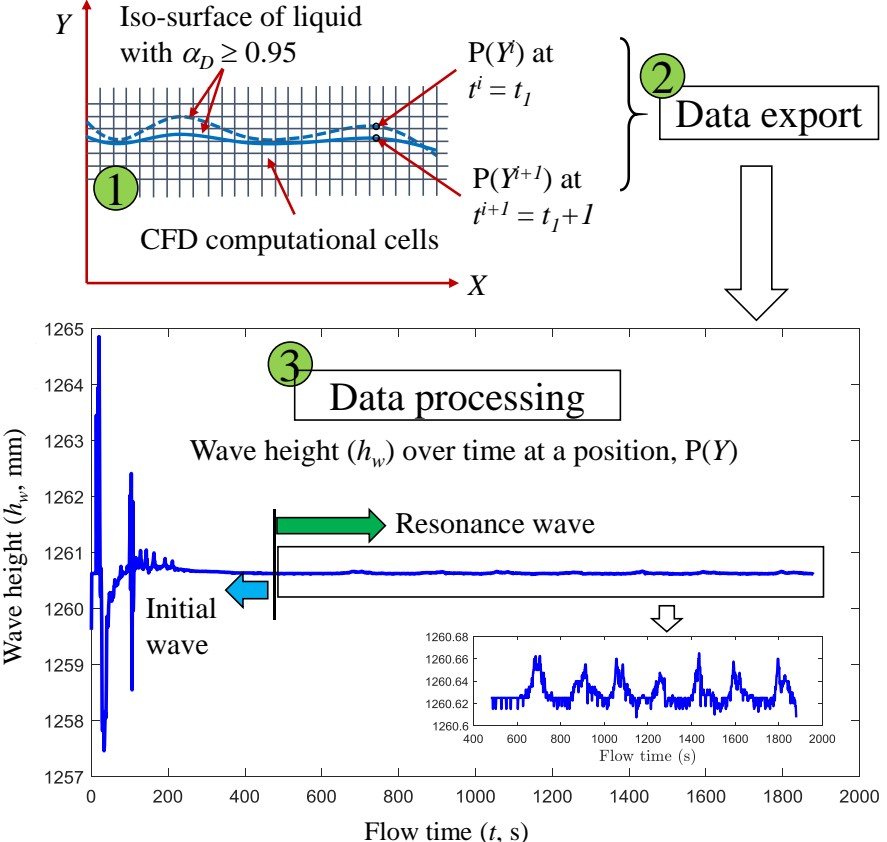

**Figure 5.** Procedure of detecting wave height from CFD results.

## 5. Results and Discussion

The three-dimensional (3D) VOF-CFD model was solved using ANSYS Fluent v18.2 (ANSYS Inc., Washington, PA, USA) and a 24-core workstation (Supermicro Inc., San Jose, CA, USA, model: X10DAi, Intel Xeon CPU E5-2670 of 2.3 GHz and 128 GB RAM). The time step of the transient simulation was fixed to 0.01 s. The calculation time was approximately five days for 30 min of the flow time in each case.

The CFD results were analyzed for the five PAN-fiber speeds ($v_{PAN}$) in terms of the streamlines of liquid flow, Froude number, wave speed on the liquid surface, and wave amplitude. The time-averaged quantities such as velocity and volume fraction were sampled at every 0.1 s for 30 min of flow time.

### 5.1. Effects of PAN Speed on Flow Streamlines

The side view of the flow streamlines on the symmetrical plane is illustrated in Figure 6 for the five $v_{PAN}$ (0.1, 0.25, 0.5, 0.75, and 1 m/s) at $t = 1800$ s. The color of the streamlines indicates the velocity magnitude, and the streamlines with the arrow describe the flow history during the residence in the bath. A streamline represents a non-intersected flow. At $v_{PAN} = 0.1$ m/s, a large recirculation flow is observed in the inner region surrounded by the PAN fiber. One recirculation flow also appears in the top-right of the bath (see Figure 6a). A recirculation flow well-developed beneath the right of the PAN outlet is observed in Figure 6e. It may be attributed from the fact that the drag force between the moving PAN fiber and the liquid promotes the recirculation along the wall of the coagulation bath as $v_{PAN}$ increases. At $v_{PAN} = 1.0$ m/s, the wake flow that emerges beneath the surface and separates the flow on the surface is observed at the PAN outlet, and in the region between the DMSO outlet and the PAN inlet.

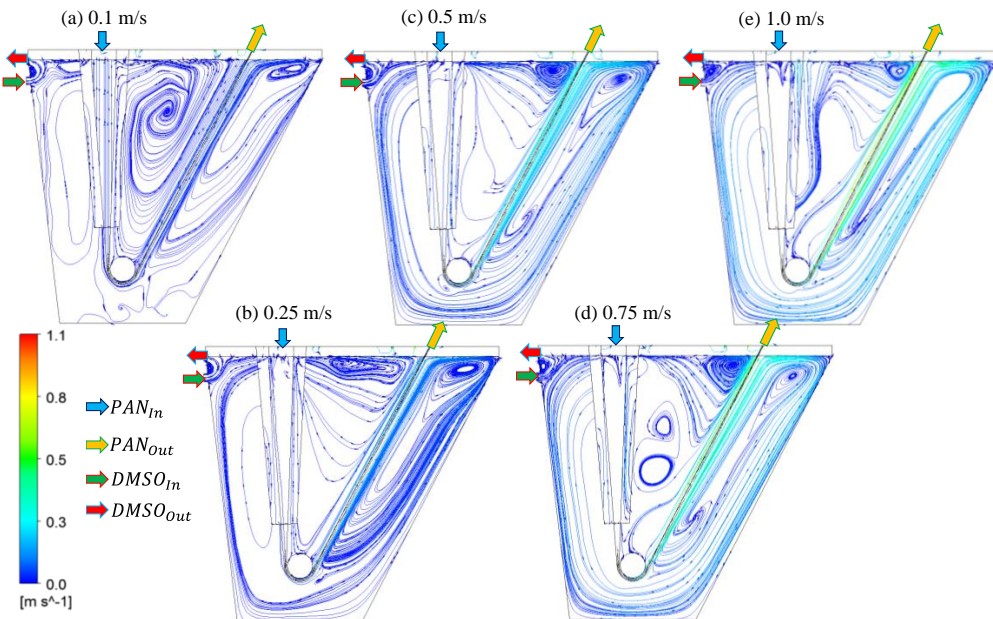

**Figure 6.** Streamlines of DMSO solution in the side-view at $t = 1800$ s.

Figure 7 depicts the streamlines from the top view of the liquid surface for the five $v_{PAN}$ at $t = 1800$ s. The two wake flows are clearly shown for each PAN speed behind the PAN inlet and at the PAN outlet. In the wake flow behind the PAN inlet, the front flow moves directly from the DMSO inlet to the outlet, while the rear flow moves forward to the PAN inlet. The wake points change according to the PAN speed. Because the forward flow from the PAN inlet collides with the backward flow from the PAN outlet, recirculation flows (or vortices) appear between the PAN inlet and PAN outlet.

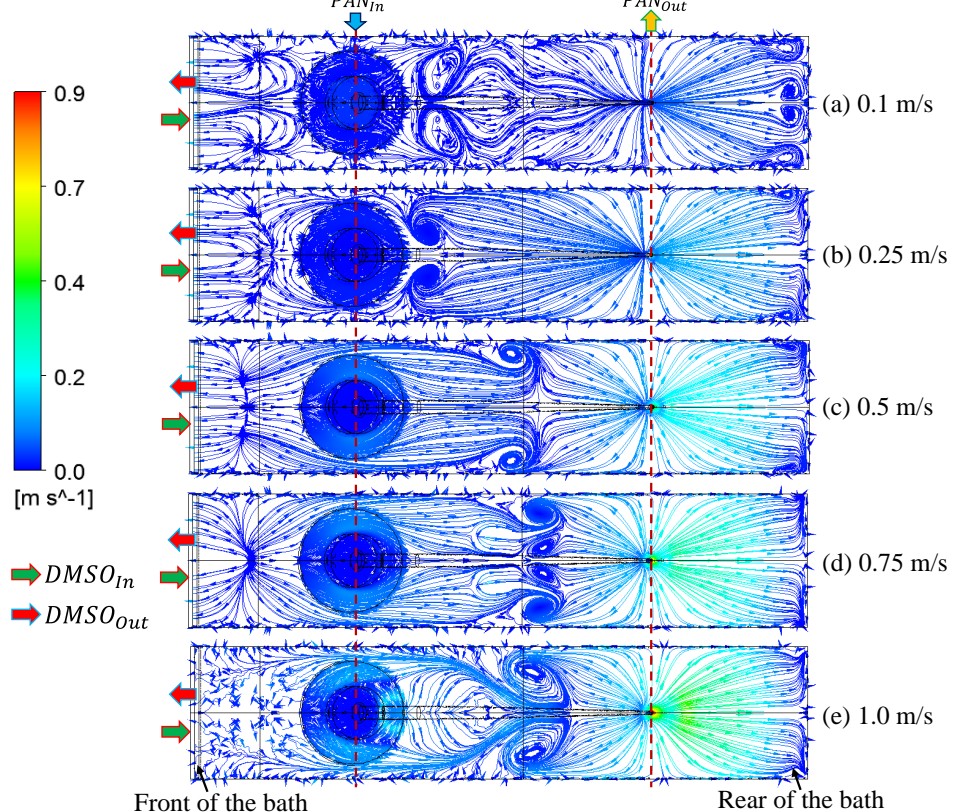

**Figure 7.** Streamlines of DMSO solution from the top view at $t = 1800s$.

## 5.2. Froude Number and Wave Speed

Figure 8 shows the contours of the time-averaged Froude number ($\overline{Fr}$) on the liquid surface at the five $v_{PAN}$. The maximum value of $\overline{Fr}$ was 0.28 near the PAN outlet at $v_{PAN} = 1$ m/s. However, the scale of $\overline{Fr}$ in Figure 8 was limited from 0 to 0.05 for an easy visualization of the difference in $\overline{Fr}$ between the five PAN speeds. Thus, red implies that $\overline{Fr}$ ranges from 0.05 to 0.28. As $\overline{Fr} < 1$ in this system, the waves can travel to both the front and rear sides of the bath.

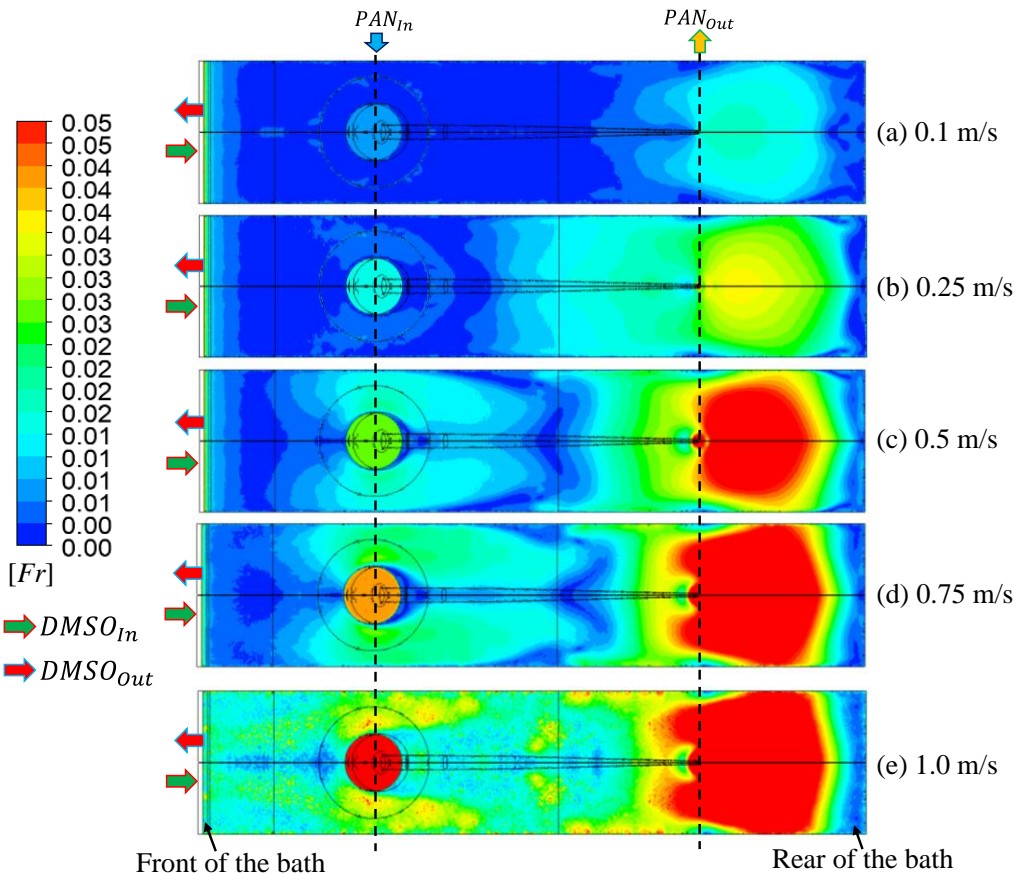

**Figure 8.** Contours of time-averaged Froude number ($\overline{Fr}$) on the liquid surface.

As explained in Equation (6), a high $\overline{Fr}$ on the liquid surface indicates a high potential for wave formation. The higher the PAN speed ($v_{PAN}$), the higher the $\overline{Fr}$ becomes. $\overline{Fr}$ is relatively high on the surfaces of the PAN inlet and the rear side of the PAN outlet. $\overline{Fr}$ is relatively low on the surfaces of the right end of the bath, the center of the bath, and behind the PAN inlet. In Figure 8d, two long ellipsoidal shapes are found near the side walls of the PAN inlet, where the wave can be formed. Some waves are observed in the region where the forward flow from the PAN inlet collides with the backward flow from the PAN outlet (see Figure 7), which is coincident with the ripples on the free surface.

Figure 9 shows the wave speed ($|u|_w$) with respect to the liquid velocity ($|u|$). The experimental data of $|u|_w$ were measured in a co-current air–water inclined-pipe flow [30]. Two CFD results were obtained for both the simple inclined-pipe flow and the present coagulation bath at $v_{PAN} = 1.0$ m/s at various positions on the liquid surface. The CFD results for the inclined-pipe flow are comparable to the experimental data, having the same order of the wave speed in magnitude. However, the difference of the wave speed between the simple inclined-pipe flow and the complex coagulation bath flow is approximately two orders of magnitude. The wave speed exhibits a linear relationship with the liquid velocity in the inclined-pipe flow [30], while the present coagulation bath showed a random response.

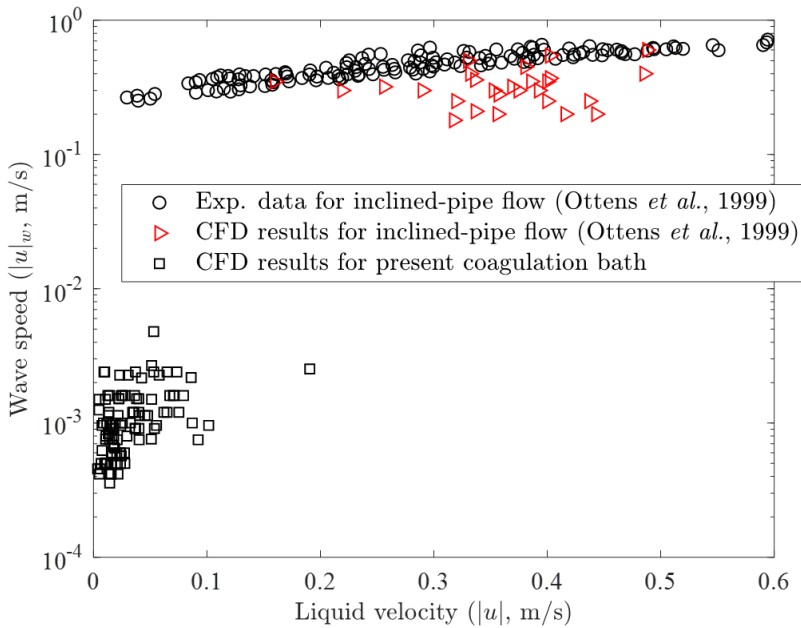

**Figure 9.** Wave speed ($|u|_w$) with respect to liquid velocity ($|u|$).

The surface wave resonance and breaking are closely related to the hydrodynamic parameters such as density, velocity, surface tension, and viscosity [31,32]. Furthermore, the geometry contributes significantly to the surface wave characteristics owing to numerous reflections from the walls and moving objects. The surface wave resonance appears somewhat chaotic in the coagulation bath, which leads to a random response of the wave speed to the liquid velocity. Therefore, it is difficult to predict the present surface wave behaviors using any previous liquid wave models [32–35].

### 5.3. Analysis of Surface Wave Signal

As mentioned in Section 4, the surface wave signal can be divided into two states: the initial state for $t \leq 480$ s and the resonance state for $t > 480$ s. Figure 10 depicts the wave height ($h_w$) at Point PF (see also Figure 4) for $v_{PAN}$ = 0.1, 0.5, 0.75, and 1.0 m/s. Because Point PF is located at the PAN inlet, the baseline of $h_w$ indicated in Figure 10b depends entirely on $v_{PAN}$. The baseline is lowered as the $v_{PAN}$ increases, as expected. For $v_{PAN}$ = 0.1 and 0.5 m/s, the initial waves are unstable and the resonance waves are stable and periodic (see Figure 10a). As shown in Figure 10b, the resonance wave at $v_{PAN}$ = 0.75 m/s is stable and periodic during the majority of the flow time, but unstable waves are occasionally superpositioned. The wave height at $v_{PAN}$ = 1.0 m/s fluctuates strongly over 480 s.

Figure 11 illustrates the MWA according to $v_{PAN}$, which was detected at ten points around the PAN inlet. For $t \leq 480$ s, the MWA is relatively high for every detecting point because of the unstable waves in the initial state (see Figure 11a). As shown in Figure 10b, the MWA of $v_{PAN}$ = 1.0 m/s is relatively lower than those of other $v_{PAN}$. This may be attributed to the fact that the strong injection of the PAN solution protects the invasion of waves into the PAN inlet area at the initial moment.

However, the MWA of $v_{PAN}$ = 1.0 m/s is the highest for $t > 480$ s for most detecting points, as shown in Figure 11b. The MWAs of $v_{PAN}$ = 0.1–0.5 m/s remain low under 0.5 mm, while the MWAs increase considerably for $v_{PAN}$ = 0.75 and 1.0 m/s. The highest value of MWA (= 8 mm) is detected at Point PF, which is located at the axial center of the PAN inlet in the forward direction. The MWA around the PAN inlet provides an air gap that is allowable according to the PAN speed.

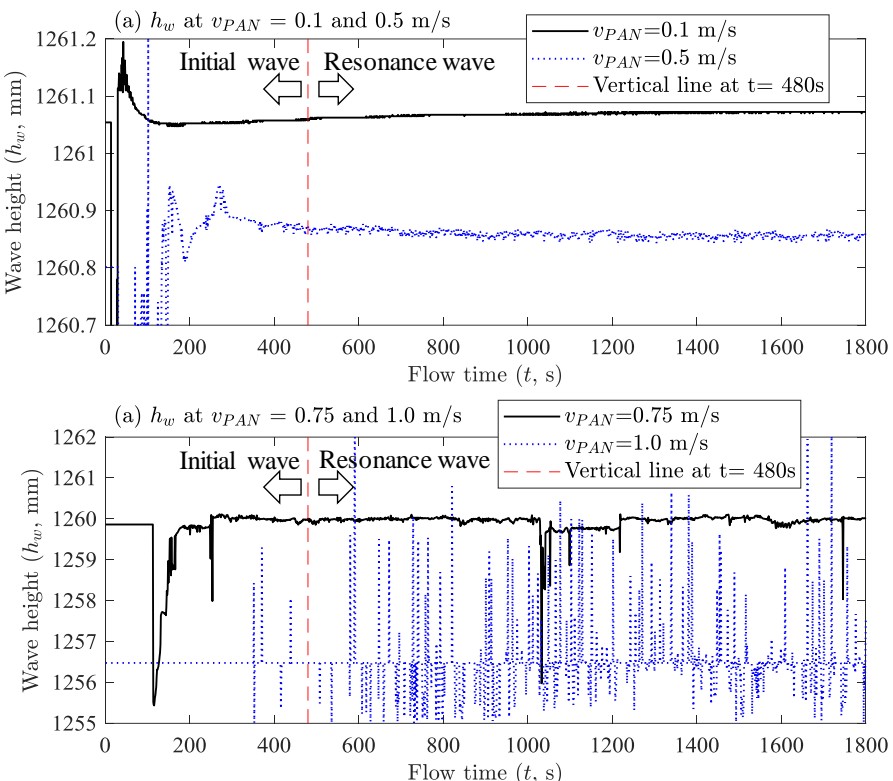

**Figure 10.** Wave height ($h_w$) from the bottom of the coagulation bath at Point PF.

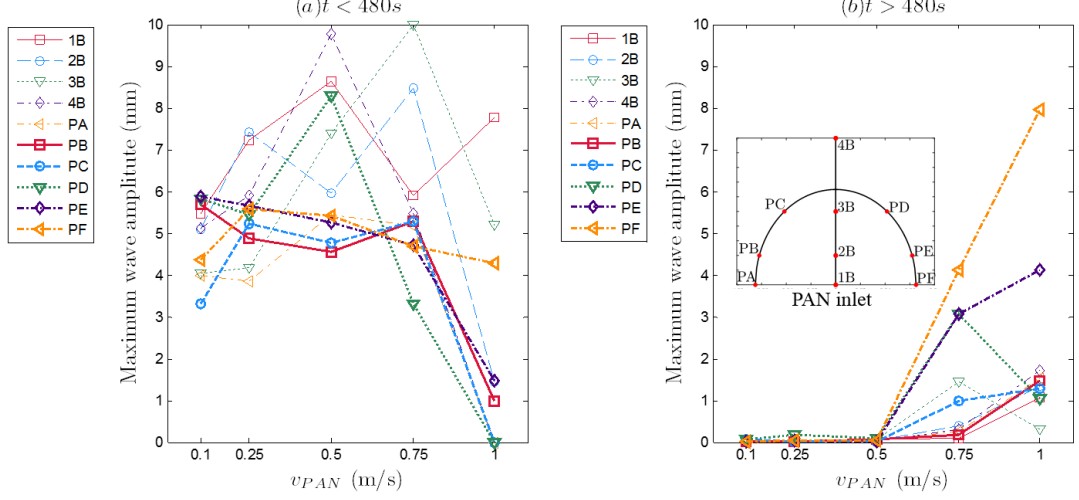

**Figure 11.** Maximum wave amplitude (MWA) around PAN inlet according to PAN speed ($v_{PAN}$).

## 6. Conclusions

In this work, a 3D VOF-CFD model was developed for a coagulation bath for the production of PAN-based CF in the DJWS process. A bundle of 3000 PAN-based CF filaments was assumed as a deformable porous zone with variations in moving velocity, porosity, and permeability along its path through the coagulation bath. The surface wave was detected at 36 points on the free surface by a three-step data processing. The iso-surface having a liquid volume fraction greater than 0.95 was tracked for 30 min of the flow time, and the wave height versus time was obtained.

The streamlines of the liquid, Froude number (*Fr*), wave speed, and wave height were examined for five PAN-fiber speeds ($v_{PAN}$ = 0.1, 0.25, 0.5, 0.75, and 1.0 m/s). The wake flow that emerged beneath the surface and separated the flow on the surface was observed in the liquid streamlines.

The contour of *Fr* indicating the potential for wave formation was coincident with the ripples on the free surface. Due to the reflections from the coagulation bath walls and moving objects, the wave speed of the coagulation bath was much lower than that of an inclined pipe flow without obstacles. The surface wave signal for 30 min was divided into the initial state ($t \leq 480$ s) and the resonance state ($t > 480$ s). The maximum wave amplitude (MWA) around the PAN inlet increased considerably for $v_{PAN}$ = 0.75 and 1.0 m/s in the resonance state. Thus, the maximum PAN production speed of 0.5 m/s was recommended for the DJWS process to maintain a stable resonance for the long-term operating condition.

The MWA can determine the minimum air gap between the spinneret and the bath surface. The maximum PAN production rate can be obtained when the air gap is close to the MWA. The resolution of the MWA depended on the minimum value of the liquid volume fraction used to determine the iso-surface. The minimum value may be confirmed by an experimental validation on the MWA. It would be useful to investigate the effect of solvent diffusion from the PAN solution on the hydrodynamics near the PAN zone to identify the reason for PAN-fiber defects during coagulation.

**Author Contributions:** Conceptualization, S.I.N. and Y.-I.L.; methodology, S.I.N.; software, S.I.N.; validation, S.I.N., Y-I.L., and S.-C.K.; formal analysis, S.I.N.; investigation, S.I.N.; resources, S.I.N.; data curation, S.I.N.; writing—original draft preparation, S.I.N.; writing—review and editing, Y.-I.L. and S.-C.K.; visualization, S.I.N.; supervision, Y.-I.L. and S.-C.K.; project administration, Y.-I.L.; funding acquisition, Y.-I.L.

**Funding:** This research was funded by the Ministry of Science, ICT, and Future Planning of Korea, grant number NRF-2016R1A2B4010423.

**Acknowledgments:** This research was supported by the Basic Science Research Program through the National Research Foundation of Korea (NRF). We acknowledge the Hyosung Research Institute for the internal communications.

**Conflicts of Interest:** The authors declare no conflict of interest. The funders had no role in the design of the study; in the collection, analyses, or interpretation of data; in the writing of the manuscript, or in the decision to publish the results.

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
