# Peer review of "Wave Characteristics of Coagulation Bath in Dry-Jet Wet-Spinning Process for Polyacrylonitrile Fiber Production Using Computational Fluid Dynamics"

_processes, doi:10.3390/pr7050314_

Round 1
Reviewer 1 Report
The spinning process is of great importance to the quality of the final carbon fiber. This work investigated the waving behavior of coagulation bath surface in during dry-jet wet-spinning of polyacrylonitrile fiber with a volume-of fluid model by carrying out computational fluid dynamics simulations. The authors presented the model and related the algorithms in detail. The wave characteristics at various spinning speeds, such as streamlines of the liquid, Froude number, wave speed, and wave height, were analyzed, and determined the maximum wave higheits. The design and structure of the work are appropriate, and the obtained results are valuable. Therefore, this work is recommended for publication. However, the following issues should be addressed before acception:
(1) The state of the roller in the bath, static or rotating, may have obvious influences on the waving behavior of the liquid phase. However, the authors did not provide such information in the manuscript. What is the state and surface properties of the roller in the model?
(2) It is stated that "The MWA provided an air gap that was allowable according to the PAN speed. For a given air 329 gap, the maximum PAN production rate could be predicted by estimating the MWA". It is not accurate. The maximum wave amplitude can only determine the minimum air gap, and only when the air gap is close to the maximum wave amplitude, it can be used to predict the maximum production rate.
(3) In Figure 9, the experimental case used to validate the model by comparison is not appropriate, since it is completely different from that investigated in this work and the calculated one presented in the Figure. As a result, the difference between the experimental and CFD results is as large as nearly three orders in magnitude. Please select another experimental case or design another calculation case for comparison. Otherwise, the comparison is of nonsense.
Author Response
The spinning process is of great importance to the quality of the final carbon fiber. This work investigated the waving behavior of coagulation bath surface in during dry-jet wet-spinning of polyacrylonitrile fiber with a volume-of fluid model by carrying out computational fluid dynamics simulations. The authors presented the model and related the algorithms in detail. The wave characteristics at various spinning speeds, such as streamlines of the liquid, Froude number, wave speed, and wave height, were analyzed, and determined the maximum wave higheits. The design and structure of the work are appropriate, and the obtained results are valuable. Therefore, this work is recommended for publication. However, the following issues should be addressed before acception:
Point 1: The state of the roller in the bath, static or rotating, may have obvious influences on the waving behavior of the liquid phase. However, the authors did not provide such information in the manuscript. What is the state and surface properties of the roller in the model?
Response 1:
Thank you for your comment.
- The following sentence in Line 71 was revised into:
<The as-spun fiber passes through a guide roller immersed in the coagulation bath.> è
<The as-spun fiber passes through a static guide roller immersed in the coagulation bath.>
- The following sentence was added to Line 73 to state the property of solid material.
<The non-slip boundary condition was applied to the surface of the roller and the wall of the bath.>
Point 2: It is stated that "The MWA provided an air gap that was allowable according to the PAN speed. For a given air gap, the maximum PAN production rate could be predicted by estimating the MWA". It is not accurate. The maximum wave amplitude can only determine the minimum air gap, and only when the air gap is close to the maximum wave amplitude, it can be used to predict the maximum production rate.
Response 2:
The sentence in Lines 339-340 was revised as follows:
<The MWA provided an air gap that was allowable according to the PAN speed. For a given air gap, the maximum PAN production rate could be predicted by estimating the MWA. > è
<The MWA can determine the minimum air gap between the spinneret and the bath surface. The maximum PAN production rate can be obtained when the air gap is close to the MWA.>
Point 3: In Figure 9, the experimental case used to validate the model by comparison is not appropriate, since it is completely different from that investigated in this work and the calculated one presented in the Figure. As a result, the difference between the experimental and CFD results is as large as nearly three orders in magnitude. Please select another experimental case or design another calculation case for comparison. Otherwise, the comparison is of nonsense.
Response 3:
Fig. 9 intents to explain the complexity and randomness of waves in the coagulation bath, compared to the simple inclined-pipe flow (see Lines 279-294). The simple inclined-pipe flow (Ottens et al., 1999) was not used for the validation of the CFD results of the coagulation tank.
To clarify this point, Fig. 9 was modified and Lines 279-287 were revised.
Figure 9. Wave speed (|u|w) with respect to liquid velocity (|u|).
è
Figure 9. Wave speed (|u|w) with respect to liquid velocity (|u|).
<Fig. 9 shows the wave speed (|u|w) with respect to the liquid velocity (|u|). The experimental data of |u|w were measured in a co-current air–water inclined-pipe flow [30]. The CFD results were obtained for vPAN = 1.0 m/s at various positions on the liquid surface. The ratio of the wave speed to the liquid velocity from the experimental data is 100 times higher than that of the current CFD results. The wave speed exhibits a linear relationship with the liquid velocity in the inclined-pipe flow [30], while the present coagulation bath showed a random response.> è
<Fig. 9 shows the wave speed (|u|w) with respect to the liquid velocity (|u|). The experimental data of |u|w were measured in a co-current air–water inclined-pipe flow [30]. Two CFD results were obtained for both the simple inclined-pipe flow and the present coagulation bath at vPAN = 1.0 m/s at various positions on the liquid surface. The CFD results for the inclined-pipe flow are comparable to the experimental data, having the same order of the wave speed in magnitude. However, the difference of the wave speed between the simple inclined-pipe flow and the complex coagulation bath flow is approximately two orders in magnitude. The wave speed exhibits a linear relationship with the liquid velocity in the inclined-pipe flow [30], while the present coagulation bath showed a random response.>

Reviewer 2 Report
This work reports on wave characteristics of coagulation bath in dry-jet wet-spinning process for polyacrylonitrile fiber production using computational fluid dynamics. It is a good piece of work and it can be considered for publication after addressing the following comments:
1. There are several reports which addressed the VOF-CFD model for the wave resonance on a free surface in the presence of obstacles and moving objects. In the present abstract, it is not clear the uniqueness of this work. Accordingly, abstract need to be revised after highlighting the unique findings of this work.
2. “As ???? increases, the drag force between the moving PAN fiber 243 and the liquid promotes the recirculation along the wall of the coagulation bath” Why? This need to be explained.
3. “The wave speed exhibits a linear relationship with the liquid velocity in the inclined-pipe flow [30], while the present coagulation bath showed a random response” Why the present coagulation bath showed a random response? This should be explained.
4. Overall quality of the present results and discussion section is average. Quality of discussion section should be improved.
Author Response
This work reports on wave characteristics of coagulation bath in dry-jet wet-spinning process for polyacrylonitrile fiber production using computational fluid dynamics. It is a good piece of work and it can be considered for publication after addressing the following comments:
Point 1: There are several reports which addressed the VOF-CFD model for the wave resonance on a free surface in the presence of obstacles and moving objects. In the present abstract, it is not clear the uniqueness of this work. Accordingly, abstract need to be revised after highlighting the unique findings of this work.
Response 1:
Thank you for your constructive comments. The abstract was revised to highlight the originality of our paper as follows:
<A three-dimensional volume-of-fluid computational fluid dynamics (VOF-CFD) model was developed for a coagulation bath of the dry-jet wet spinning (DJWS) process for the production of polyacrylonitrile (PAN)-based carbon fiber. The wave amplitude over time was detected from the surface wave having a liquid volume fraction of 0.95. The wave signal for 30 min was divided into the initial and resonance states. The maximum wave amplitude was less than 0.5 mm around the PAN-fiber inlet nozzle for vPAN = 0.1–0.5 m/s. The VOF-CFD model is useful to determine the maximum vPAN under an allowable air gap of the DJWS process> è
<A three-dimensional volume-of-fluid computational fluid dynamics (VOF-CFD) model was developed for a coagulation bath of the dry-jet wet spinning (DJWS) process for the production of polyacrylonitrile (PAN)-based carbon fiber. The PAN-fiber was assumed as a deformable porous zone with variations in moving speed, porosity, and permeability. The Froude number interpreted as the wave-making resistance on the liquid surface was analyzed according to the PAN-fiber wind-up speed (vPAN). The wave amplitude over time was detected from the surface wave having a liquid volume fraction of 0.95. The wave signal for 30 min was divided into the initial and resonance states that were distinguished at 8 min. The maximum wave amplitude was less than 0.5 mm around the PAN-fiber inlet nozzle for vPAN = 0.1–0.5 m/s in the resonance state. The VOF-CFD model is useful to determine the maximum vPAN under an allowable air gap of the DJWS process.>
Point 2: “As ???? increases, the drag force between the moving PAN fiber 243 and the liquid promotes the recirculation along the wall of the coagulation bath” Why? This need to be explained.
Response 2:
The sentence in Lines 246-249 was revised:
< As vPAN increases, the drag force between the moving PAN fiber and the liquid promotes the recirculation along the wall of the coagulation bath.> è
<A recirculation flow well-developed beneath the right of the PAN outlet is observed in Fig. 6e. It may be attributed from the fact that the drag force between the moving PAN fiber and the liquid promotes the recirculation along the wall of the coagulation bath as vPAN increases.>
Point 3: “The wave speed exhibits a linear relationship with the liquid velocity in the inclined-pipe flow [30], while the present coagulation bath showed a random response” Why the present coagulation bath showed a random response? This should be explained.
Response3:
The reason why the present coagulation bath showed a random response was explained in Lines 279-284 in the previous manuscript. To clarify the reason of the random response, Lines 288-294 was revised as follows:
<The surface wave resonance and breaking are closely related to the hydrodynamic parameters such as density, velocity, surface tension, and viscosity [31, 32]. Furthermore, the geometry contributes significantly to the surface wave characteristics owing to numerous reflections from the walls and moving objects. The surface wave resonance appears somewhat chaotic in the coagulation bath. Therefore, it is difficult to predict the present surface wave behaviors using any previous liquid wave models [32-35]>. è
<The surface wave resonance and breaking are closely related to the hydrodynamic parameters such as density, velocity, surface tension, and viscosity [31, 32]. Furthermore, the geometry contributes significantly to the surface wave characteristics owing to numerous reflections from the walls and moving objects. The surface wave resonance appears somewhat chaotic in the coagulation bath, which leads to a random response of the wave speed to the liquid velocity. Therefore, it is difficult to predict the present surface wave behaviors using any previous liquid wave models [32-35]>.
Point 4: Overall quality of the present results and discussion section is average. Quality of discussion section should be improved.
Response 4:
Thank you for your comments. Fig. 9 was revised and the explanation was added.
Figure 9. Wave speed (|u|w) with respect to liquid velocity (|u|).
<Fig. 9 shows the wave speed (|u|w) with respect to the liquid velocity (|u|). The experimental data of |u|w were measured in a co-current air–water inclined-pipe flow [30]. Two CFD results were obtained for both the simple inclined-pipe flow and the present coagulation bath at vPAN = 1.0 m/s at various positions on the liquid surface. The CFD results for the inclined-pipe flow are comparable to the experimental data, having the same order of the wave speed in magnitude. However, the difference of the wave speed between the simple inclined-pipe flow and the complex coagulation bath flow is approximately two orders in magnitude. The wave speed exhibits a linear relationship with the liquid velocity in the inclined-pipe flow [30], while the present coagulation bath showed a random response.>
